# Mammals adjust diel activity across gradients of urbanization

Travis Gallo[1,2]*, Mason Fidino[2], Brian Gerber[3], Adam A Ahlers[4], Julia L Angstmann[5], Max Amaya[6], Amy L Concilio[7], David Drake[8], Danielle Gay[9], Elizabeth W Lehrer[2], Maureen H Murray[2], Travis J Ryan[5], Colleen Cassady St Clair[10], Carmen M Salsbury[5], Heather A Sander[11], Theodore Stankowich[6], Jaque Williamson[12], J Amy Belaire[13], Kelly Simon[14], Seth B Magle[2]

[1]Environmental Science and Policy, College of Science, George Mason University, Fairfax, United States; [2]Urban Wildlife Institute, Conservation and Science Department, Lincoln Park Zoo, Chicago, United States; [3]Department of Natural Resource Science, The University of Rhode Island, Kingston, United States; [4]Department of Horticulture and Natural Resources, Kansas State University, Manhattan, United States; [5]Department of Biological Sciences and Center for Urban Ecology and Sustainability, Butler University, Indianapolis, United States; [6]Department of Biological Sciences, California State University Long Beach, Long Beach, United States; [7]Department of Environmental Science and Policy, St. Edward's University, Austin, United States; [8]Department of Forest and Wildlife Ecology, University of Wisconsin-Madison, Madison, United States; [9]Austin Parks and Recreation, City of Austin, Austin, United States; [10]Department of Biological Sciences, University of Alberta, Edmonton, Canada; [11]Department of Geographical and Sustainability Sciences, University of Iowa, Iowa City, United States; [12]Department of Education & Conservation, Brandywine Zoo, Wilmington, United States; [13]The Nature Conservancy in Texas, San Antonio, United States; [14]Texas Parks and Wildlife, Austin, United States

*For correspondence:
hgallo@gmu.edu

**Competing interest:** The authors declare that no competing interests exist.

**Abstract** Time is a fundamental component of ecological processes. How animal behavior changes over time has been explored through well-known ecological theories like niche partitioning and predator–prey dynamics. Yet, changes in animal behavior within the shorter 24-hr light–dark cycle have largely gone unstudied. Understanding if an animal can adjust their temporal activity to mitigate or adapt to environmental change has become a recent topic of discussion and is important for effective wildlife management and conservation. While spatial habitat is a fundamental consideration in wildlife management and conservation, temporal habitat is often ignored. We formulated a temporal resource selection model to quantify the diel behavior of 8 mammal species across 10 US cities. We found high variability in diel activity patterns within and among species and species-specific correlations between diel activity and human population density, impervious land cover, available greenspace, vegetation cover, and mean daily temperature. We also found that some species may modulate temporal behaviors to manage both natural and anthropogenic risks. Our results highlight the complexity with which temporal activity patterns interact with local environmental characteristics, and suggest that urban mammals may use time along the 24-hr cycle to reduce risk, adapt, and therefore persist, and in some cases thrive, in human-dominated ecosystems.

## Editor's evaluation

This study will be of interest to wildlife ecologists and conservation practitioners. The authors took a collaborative approach and collated a large dataset of wildlife camera trap recordings across cities in the USA. The analyses reveal variability in diel activity among species and cities, providing important insights into the effects of urbanization.

## Introduction

Time is a fundamental axis that shapes ecological systems. Regarding animal behavior, time, and space are linked in that the spatial characteristics of an animal's local environment influences its temporal behavior (*Kronfeld-Schor and Dayan, 2003*). For example, some species make fine-scale adjustments to their temporal behavior to respond to predation risk (*van der Vinne et al., 2019*), competition (*Kronfeld-Schor and Dayan, 2003*), food availability (*Owen-Smith, 2008*), seasonal variability in local climatic conditions (*Maloney et al., 2005*), and even lunar cycles (*Prugh and Golden, 2014*). While temporal behavior has yet to become a major focus in animal ecology (*Gaston, 2019*; *Kronfeld-Schor and Dayan, 2003*), how animals use time as an ecological resource has inspired well-known ecological phenomenon like niche partitioning (*Schoener, 1974*) and predator–prey dynamics (*Tambling et al., 2015*). From an applied perspective, understanding if an animal can make temporal adjustments to mitigate or adapt to local environmental change remains a topic of discussion (*Wolkovich et al., 2014*), and is important for effective wildlife management and conservation (*Levy et al., 2019*).

Species that persist in human-dominated environments, like cities, require some degree of human avoidance to safely navigate these complex landscapes (*Gehrt et al., 2009*; *Murray and St. Clair, 2015*; *Riley et al., 2003*). In urban ecosystems, few habitat patches remain for animals to seek spatial refuge when confronted with human disturbance and/or negative interactions with other species. In these cases, temporally partitioning from these potentially dangerous interactions might be an alternative strategy. A recent global meta-analysis suggests that mammals become more nocturnal in areas with greater human disturbance (*Gaynor et al., 2018*). However, *Frey et al., 2020* found that shifts in temporal behavior of apex predators – in response to human disturbance – caused cascading behavioral responses among mesocarnivores creating a 'behavioral release'. These results highlight that temporal shifts toward nocturnality are not universal and may be context specific.

Of the 76 studies assessed in the *Gaynor et al., 2018* global meta-analysis, only 7.8% ($n = 11$) assessed changes in nocturnal activity in urban areas, and all explored these changes categorically between urban and nonurban areas. Binary urban and rural categorizations generally fail to capture variation in urban development and cannot generate generalizable results that correlate to other cities (*McDonnell and Pickett, 1990*). Additionally, cities are unique and differ in size, land use, growth patterns, and human culture (*Pacione, 2009*). Variation in both spatial and temporal characteristics within and among cities could have differing effects on animal behavior. Thus, key questions remain regarding the way in which animal diel activity varies across gradients of urbanization and among differing cities. For example, the magnitude of change in diel activity patterns may be larger for more densely urbanized cities or may depend on regional variation in day and nighttime temperatures. Multicity investigations that include variation in urban intensity and regional climate can elucidate such patterns.

*Gaynor et al., 2018* found that most studies in urban environments also focused on carnivore species, highlighting a gap in our understanding regarding changes in diel activity across taxa. For example, carnivores likely avoid humans in both space and time because of inimical human interactions (*Clinchy et al., 2016*; *Kitchen et al., 2000*). This may not be the case for mammals that do not regularly come in conflict with humans or do not evoke such visceral reactions by humans. Additionally, some species may be constrained by their morphology (e.g., number and type of cones and rods in their eyes) or may otherwise lack the ability to be active in differing light levels. To fully understand the variability of activity patterns and assess temporal adjustments in response to urban development, a comprehensive examination of the larger suite of urban mammals and across multiple urban environments is required.

While spatial habitat is a fundamental consideration in wildlife management and conservation, temporal habitat is often ignored (*Gaston, 2019*). Here, we link spatial landscape characteristics with the diel activity patterns of 8 terrestrial mammals using remote cameras deployed across 10 US cities.

**Table 1.** The total number of detections for each species, number of cities each species was detected in, mean proportion of sites each species was detected at per city, and total number of detections in each time category for 8 urban mammal species across 10 US metropolitan areas between January 2017 and December 2018.

| Species | Total detections | No. of cities species detected | Mean proportion of sites species detected per city | No. of 'day' detections | No. of 'dawn' detections | No. of 'dusk' detections | No. of 'night' detections | No. of 'deep night' detections |
|---|---|---|---|---|---|---|---|---|
| Bobcat | 102 | 5 | 0.16 | 29 | 1 | 9 | 45 | 18 |
| Coyote | 2732 | 9 | 0.63 | 671 | 98 | 256 | 1318 | 389 |
| Eastern cottontail | 16,102 | 10 | 0.61 | 3984 | 619 | 1097 | 8317 | 2085 |
| Raccoon | 34,931 | 10 | 0.77 | 2638 | 642 | 3767 | 21,723 | 6161 |
| Red fox | 1570 | 8 | 0.51 | 441 | 35 | 152 | 744 | 198 |
| Striped skunk | 990 | 10 | 0.24 | 89 | 24 | 98 | 584 | 195 |
| Virginia opossum | 8357 | 8 | 0.7 | 407 | 116 | 1027 | 5087 | 1720 |
| White-tailed deer | 14,875 | 10 | 0.56 | 7965 | 658 | 816 | 4299 | 1137 |

Our research objective was to determine which species change their diel activity across gradients of urbanization and identify what characteristics of the urban environments have the strongest association with changes in diel activity.

We found high variability in diel activity patterns within and among species and species-specific correlations between diel activity and human population density, impervious land cover, available greenspace, vegetation cover, and mean daily temperature. Our results indicate that in high-risk environments, such as cities, animals may reduce risk by modulating their temporal habitat use. Our study identifies a potential mechanism by which urban wildlife species may adapt to human-dominated environments, and provides critical insight into activity patterns of urban wildlife that will prove useful for managing these species in cities.

## Results

To quantify changes in mammal diel activity in response to urbanization, we used camera detection data for eight common urban mammal species: bobcat (*Lynx rufus*), coyote (*Canis latrans*), red fox (*Vulpus vulpus*), raccoon (*Procyon lotor*), striped skunk (*Mephitis mephitis*), eastern cottontail (*Sylvilagus floridanus*), Virginia opossum (*Didelphis virginiana*), and white-tailed deer (*Odocoileus virginianus*). Cameras were deployed in a systematic fashion across 10 US metropolitan areas as part of the Urban Wildlife Information Network (https://urbanwildlifeinfo.org/): Austin, Texas, Chicago, Illinois, Denver and Fort Collins, Colorado, Indianapolis, Indiana, Iowa City, Iowa, Orange County, California, Madison, Wisconsin, Manhattan, Kansas, and Wilmington, Delaware (*Appendix 1—figure 1*; see *Fidino et al., 2021*; *Magle et al., 2019* for details).

Across 41,594 trap nights (*Supplementary file 1a*), we captured 79,659 total unique detection events. Total detections per species ranged from 102 to 34,931, and each species was detected in 5–10 cities at an average proportion of 0.16–0.77 sites per city (*Table 1*). Bobcat occurred at the lowest number of cities and proportion of sites, while raccoon occurred in all 10 cities and at the greatest proportion of sites (*Table 1*, see *Supplementary file 1b* for the proportion of sites in each city). The number of detections captured throughout the 24-hr diel period varied among species (*Table 1*).

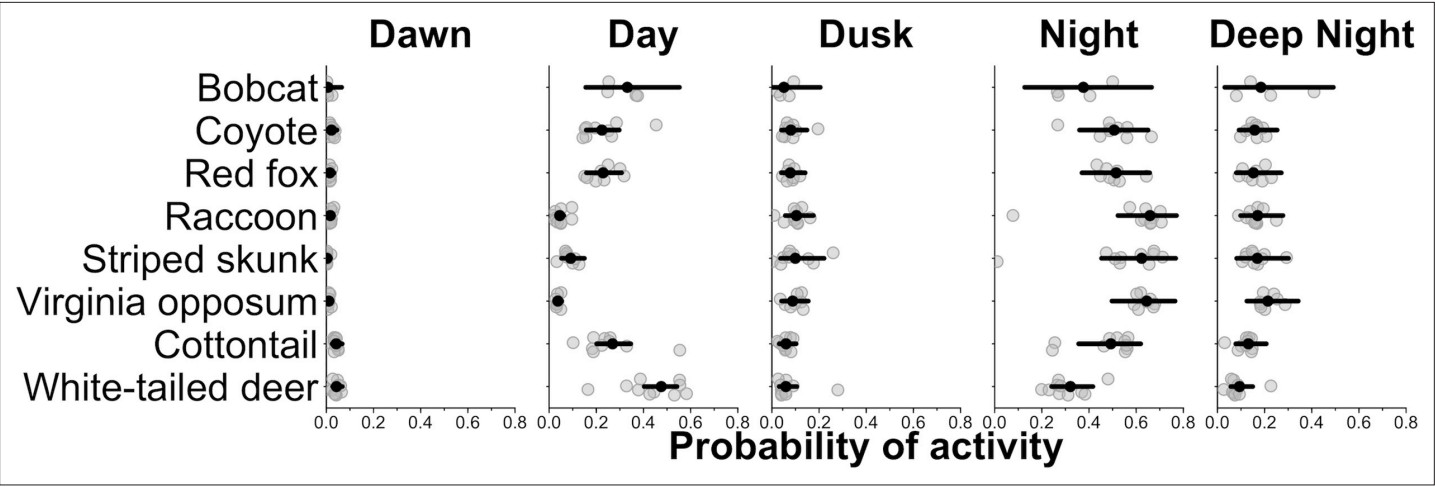

**Figure 1.** City-specific probability of activity for each species. Gray points are city-specific estimates of the average probability of activity in each time category. The black point indicates the average probability of activity among cities and the horizontal lines are 95% credible interval for the average probability estimates among cities. Wider credible intervals indicate more variation among cities.

## Modeling diel activity

We formulated a hierarchical multinomial model to quantify the diel behavior of each species and assess the effects that available greenspace, vegetation cover, impervious land cover, human population density, and daily temperature had on diel behavior of each species. Our approach operates similar to resource selection functions in which resources are selected in space. However, substituting time for space allowed us to quantify changes in diel activity across gradients of environmental change. This temporal resource selection model allowed us to estimate temporal 'selection' and the probability of 'use' in each time category. Coefficient estimates are estimates of selection for a particular time category relative to the available time in the respective category and the difference from the reference time category ('day'). Exponentiated coefficient estimates greater than one indicates selection and less than one indicates avoidance, relative to the day reference category. We also estimated the influence that each predictor variable had on the probability of activity in each time category, including the 'day' category, using the softmax function – a generalization of the inverse logit link for more than two modeled categories (*Kruschke, 2011*).

## Among city variation in diel activity patterns

We found that most species, on average, had a higher probability of being nocturnal (active at night or during the darkest portions of night) with the exception of bobcat and white-tailed deer (*Figures 1 and 2*). Most species showed variation in diel activity among cities (e.g., bobcat; *Figure 1*), and some species (e.g., eastern cottontail, coyote, red fox, and bobcat) exhibited profound variation in diel activity across individual sampling sites (*Figure 2*). For example, the predicted probability of nocturnal behavior for eastern cottontail at each sampled site ranged from 0.15 to 0.69 (see *Supplementary file 1c* for a full set of ranges for each species and each time category).

## Selection for particular time categories

Of the three predator species that we analyzed (coyote, bobcat, and red fox), we found that anthropogenic and natural features were associated with variation in diel activity for only coyote and red fox (*Figure 3A-C*). Coyote selected for both nocturnal and crepuscular hours in areas of greater human population densities (*Figure 3B*), and red fox avoided nocturnal hours in areas with more available greenspace (*Figure 3C*). Seasonality also had an effect on both coyote and fox diel activity. Coyote selected for dawn hours (*Figure 3B*) and red foxes selected for dusk hours during periods of higher daily average temperatures (*Figure 3C*). We found no evidence that bobcats varied their diel activity across our environmental variables (*Figure 3A*).

We found diel activity for all omnivore and herbivore species was affected by anthropogenic features. Raccoon, eastern cottontail, and white-tailed deer avoided nighttime hours in areas of

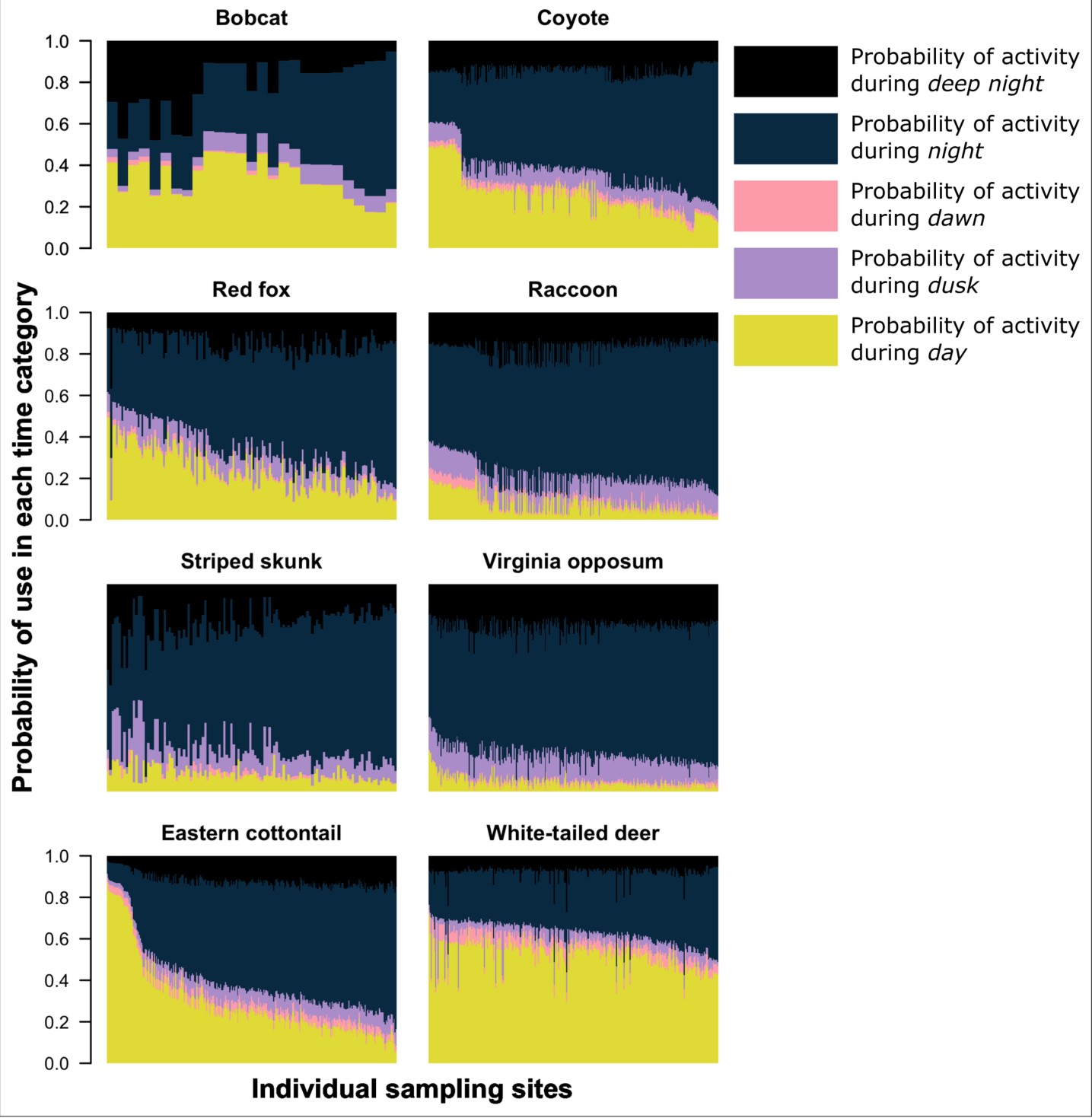

**Figure 2.** The predicted probability of activity in each time category at each sampling site (*x*-axis) the species was detected. Each column on the *x*-axis is a stacked bar plot representing the probability of activity in each time category at each sampling site. For each bar plot, all categories sum to one. Sampling sites along the *x*-axis are ordered from the lowest probability of nocturnal activity to the highest.

greater human population density (*Figure 3D, G, H*), whereas Virginia opossum selected for nighttime and dusk hours in areas with greater human densities (*Figure 3F*). Raccoon, striped skunk, and white-tailed deer all selected for nighttime hours in areas with greater impervious land cover (*Figure 3D, E, H*).

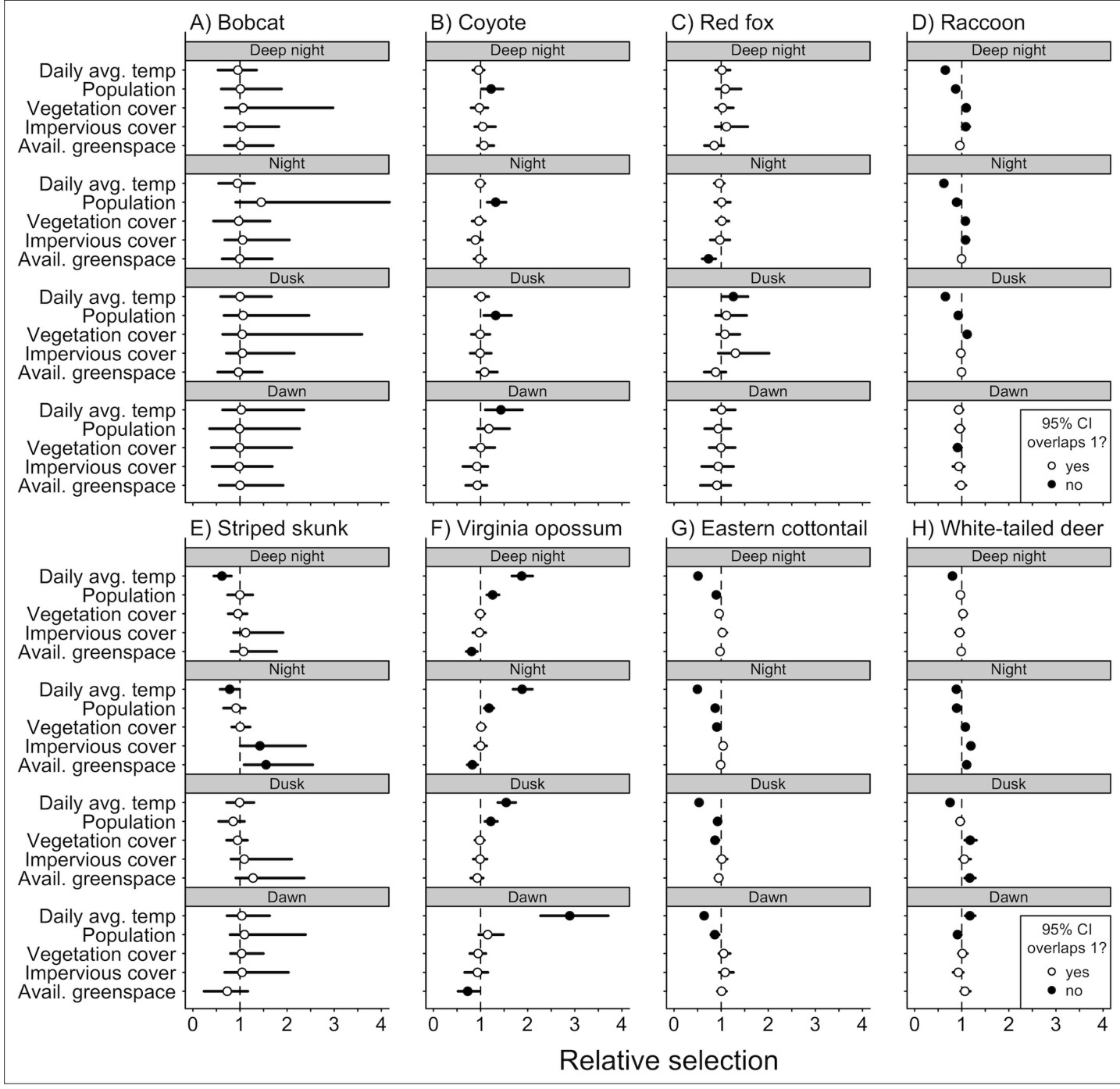

**Figure 3.** Mean (circle) and 95% credible intervals of estimated coefficients from natural and anthropogenic features on temporal selection of deep night, night, dusk, and dawn relative to day.

Natural features were also associated with variation in diel activity for omnivore and herbivore species. As vegetation cover increased, eastern cottontails were more likely to select daytime hours (*Figure 3G*), whereas raccoons and white-tailed deer were more likely to select for nighttime hours and dusk (*Figure 3D, H*). As available greenspace increased, striped skunk were more likely to select nighttime hours (*Figure 3E*), whereas Virginia opossum were less likely to select nighttime and dawn hours (*Figure 3F*). White-tailed deer were also more likely to select nighttime and dusk hours as available greenspace increased (*Figure 3H*).

We found seasonality effects on all omnivore and herbivore species. Virginia opossum were more likely to avoid daytime hours as temperatures increased (*Figure 3F*). Daily average temperature had a positive relationship with diurnal selection for raccoons, striped skunk, eastern cottontail, and white-tailed deer (*Figure 3D, E, G, H*).

## Probability of nocturnal activity

To further assess the probability of a species shifting toward nocturnality as a response to natural and anthropogenic features of the environment, we combined the probability of activity during night and deep night and predicted these values across each continuous covariates using the model results for

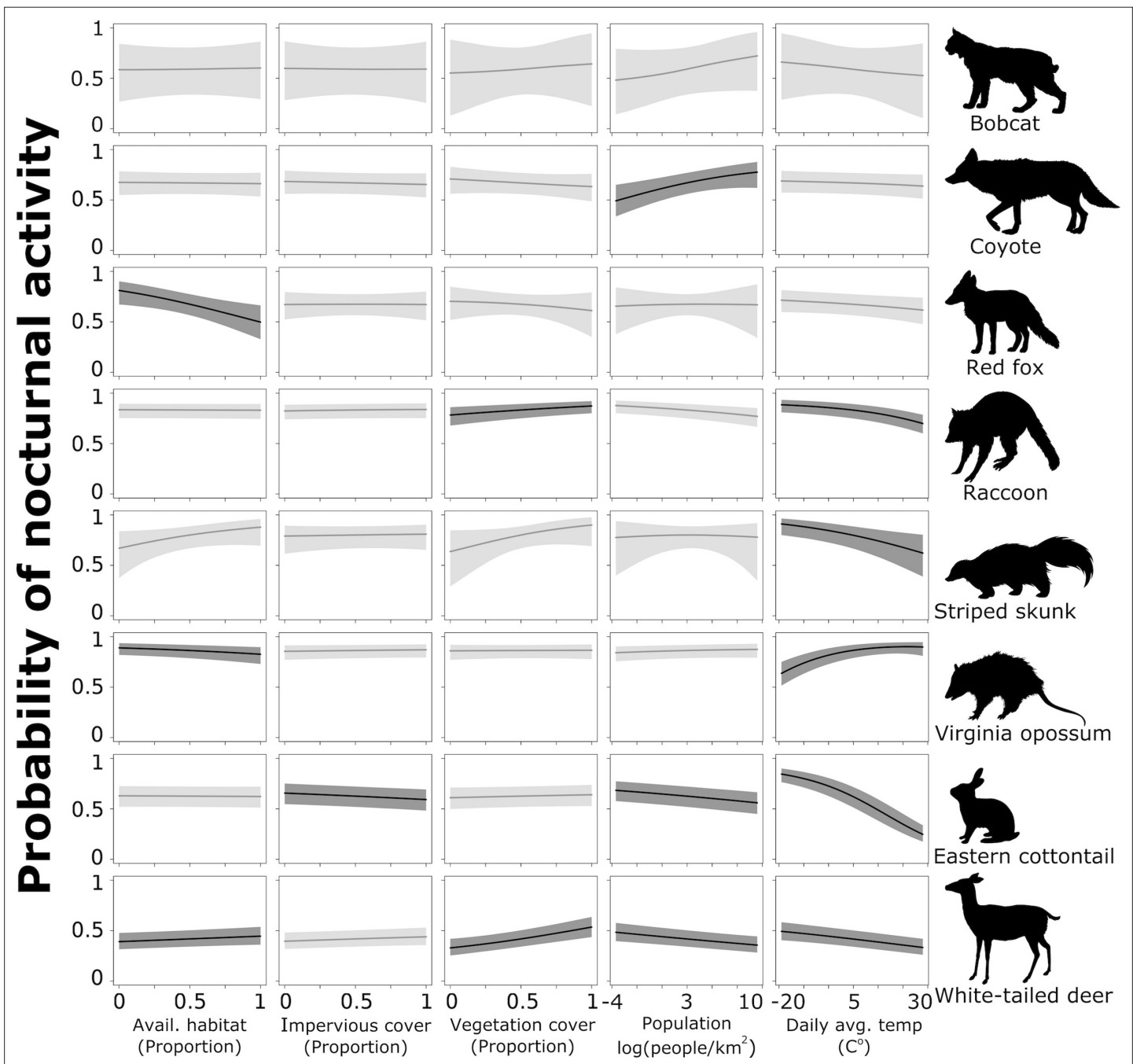

**Figure 4.** Probability of nocturnal activity (night and deep night combined) across each of our natural and anthropogenic characteristics of the urban environment. Solid line indicates the median predicted line and shaded areas are 95% credible interval. Darker shading represent the relationships whose odds ratios did not overlap 1.

each species. Coyote had a lower probability of being nocturnal in areas with lower human densities, but that probability increased significantly as human population increased (*Figure 4*). With a 1 standard deviation (hereafter sd) increase from the mean human population density (from 1512 to 3095 people/km$^2$), coyotes are 18% more likely to use nighttime hours and 38% more likely with a 2 sd increase from 1512 to 4678 people/km$^2$ (*Table 2*). Across the available habitat gradient red fox were 21% less likely to use nighttime hours with a 1 sd increase in available greenspace from 0.41 to 0.57, and 38% less likely with a sd increase from 0.41 to 0.73 (*Table 2*). Note that predictor values vary because they were collected at species-specific scales and not all species were detected at the same sites.

White-tailed deer, eastern cottontail, and raccoon had a greater probability of being active at night where human densities were low; this probability decreased as human population increased (*Figure 4*). White-tailed deer were 8% less likely to use nighttime hours with a 1 sd increase in population density from 1515 to 3003 people/km$^2$, eastern cottontail were 10% less likely (from 2226 to 4633 people/km$^2$), and raccoon were 7% less likely (from 1763 to 3789 people/km$^2$; *Table 2*). With a 2 sd increase in population density (1515–4491 people/km$^2$ for white-tailed deer, 2226–7040 for eastern cottontail, and 1763–5815 for raccoon), white-tailed deer were 15% less likely to be nocturnal, eastern cottontail 19% less likely, and raccoon 13% less likely to be nocturnal (*Table 2*). Conversely, white-tailed deer and raccoon showed a positive relationship with increased impervious cover and nocturnality (*Figure 4*). White-tailed deer were 14% more likely to be active at night with a 1 sd increase in impervious cover from 0.16 to 0.31% and 31% more likely with a 2 sd increase from 0.16 to 0.45 (*Table 2*). Raccoons were 10% more likely to be active at night with a 1 sd increase in impervious cover and 21% more likely with an a 2 sd increase (*Table 2*).

Vegetation cover had a negative effect on the probability of nocturnal behavior of eastern cottontail (*Figure 4*). Cottontail were 7% less likely to be nocturnal when the proportion of vegetation cover increased 1 sd above the mean from 0.67 to 0.92, and 13% less likely to be nocturnal when vegetation cover increased 2 sd above the mean from 0.67 to 1 (*Table 2*). We also found that white-tailed deer were 5% more likely to use nighttime hours when the proportion of available greenspace increased 1 sd above the mean from 0.52 to 0.75% and 11% more likely with an increase of 2 sd from the mean from 0.52 to 0.98 (*Table 2*). However, Virginia opossum were 12% less likely to be nocturnal with a 1 sd increase in available greenspace from 0.34 to 0.57% and 23% less likely with an increase of 2 sd from 0.34 to 0.78 (*Table 2*).

Finally, we found an influence of daily average temperature (season) on eastern cottontail, raccoon, striped skunk, white-tailed deer, and Virginia opossum (*Figure 4*). Eastern cottontail were 43% less likely to use nighttime hours with a 1 sd increase in daily average temperature from 8.17 to 18.76°C, and 69% less likely with a 2 sd increase from 8.17 to 29.36°C (*Table 2*). With a 1 sd increase in temperature from 12.00 to 21.49°C, raccoon were 19% less likely to use nighttime hours, and 37% less likely with a 2 sd increase from 12.00 to 30.99°C (*Table 2*). Striped skunk were 25% less likely to exhibit nocturnal behavior with a 1 sd increase in daily average temperature from 15.3 to 24.62°C, and 44% less with a 2 sd increase from 15.3 to 33.94°C (*Table 2*). White-tailed deer were 12% less likely with a 1 sd increase from 12.01 to 22.65°C and 23% less likely with a 2 sd increase from 12.01 to 33.30°C (*Table 2*). Virginia opossum, however, were 25% more likely to use nighttime hours with a 1 sd increase in daily average temperature from 13.85 to 22.28°C, and 42% more likely with a 2 sd increase from 13.85 to 30.71°C (*Table 2*). Again, temperature ranges vary because not all species were detected at the same sites and same times.

## Discussion

Ecological processes act across both space and time. We have, however, only just begun to study how animals use diel time as an ecological resource to avoid risk and adapt to environmental change. We quantified the diel behavior of 8 mammal species across urban gradients in 10 US cities. Our findings indicated that mammals can modulate their use of time within the 24-hr diel period as a resource to persist in urban ecosystems. We found that nocturnal activity had the greatest response to urbanization and seasonality, and that changes in nocturnality in response to urbanization were species specific and varied among cities. Our results also illustrated the complex trade-offs that urban wildlife species must make to contend with both interspecific interactions (i.e., predation and competition)

**Table 2.** Odds ratios for each predictor variable and a 1 and 2 standard deviation increase across their values. Bolded text indicates scenarios where the 95% credible intervals do not overlap 1.

| | Available greenspace | | Impervious cover | | Vegetation cover | | Human pop. density | | Daily avg. temp | |
|---|---|---|---|---|---|---|---|---|---|---|
| | 1-Unit increase | 2-Unit increase | 1-Unit increase | 2-Unit increase | 1-Unit increase | 2-Unit increase | 1-Unit increase | 2-Unit increase | 1-Unit increase | 2-Unit increase |
| Bobcat | 1.01 | 1.03 | 1.04 | 1.09 | 1.00 | 1.00 | 1.27 | 1.64 | 0.94 | 0.89 |
| | (0.72–1.58) | (0.52–2.58) | (0.74–1.75) | (0.54–3.18) | (0.57–1.77) | (0.30–3.51) | (0.87–3.09) | (0.74–10.4) | (0.59–1.24) | (0.35–1.56) |
| Coyote | 0.99 | 0.97 | 0.94 | 0.89 | 0.97 | 0.94 | **1.18** | **1.38** | 0.95 | 0.89 |
| | (0.88–1.10) | (0.76–1.22) | (0.80–1.09) | (0.64–1.19) | (0.84–1.09) | (0.71–1.18) | (1.04–1.35) | (1.05–1.80) | (0.87–1.03) | (0.74–1.06) |
| Red fox | **0.79** | **0.62** | 0.94 | 0.87 | 1.00 | 0.99 | 1.00 | 1.00 | 0.92 | 0.83 |
| | (0.66–0.93) | (0.44–0.86) | (0.75–1.14) | (0.53–1.30) | (0.89–1.13) | (0.78–1.27) | (0.86–1.17) | (0.73–1.38) | (0.82–1.01) | (0.66–1.02) |
| Raccoon | 0.99 | 0.99 | **1.10** | **1.20** | 1.02 | 1.03 | **0.93** | **0.87** | **0.81** | **0.63** |
| | (0.95–1.03) | (0.91–1.07) | (1.05–1.16) | (1.10–1.34) | (0.98–1.06) | (0.96–1.11) | (0.90–0.97) | (0.81–0.94) | (0.76–0.85) | (0.55–0.70) |
| Striped skunk | 1.26 | 1.58 | 1.27 | 1.60 | 1.02 | 1.04 | 1.00 | 0.99 | **0.75** | **0.56** |
| | (0.93–1.77) | (0.84–3.09) | (0.93–1.87) | (0.85–3.48) | (0.86–1.22) | (0.73–1.49) | (0.79–1.22) | (0.60–1.46) | (0.59–0.92) | (0.35–0.84) |
| Virginia opossum | **0.88** | **0.77** | 1.01 | 1.02 | 1.03 | 1.05 | 1.04 | 1.07 | **1.25** | **1.42** |
| | (0.81–0.96) | (0.65–0.92) | (0.91–1.12) | (0.82–1.24) | (0.96–1.10) | (0.92–1.20) | (0.97–1.11) | (0.93–1.23) | (1.14–1.36) | (1.10–1.70) |
| Eastern cottontail | 0.99 | 0.99 | 1.03 | 1.05 | **0.93** | **0.87** | **0.90** | **0.81** | **0.57** | **0.31** |
| | (0.95–1.04) | (0.90–1.07) | (0.97–1.09) | (0.94–1.18) | (0.89–0.98) | (0.78–0.96) | (0.86–0.94) | (0.75–0.89) | (0.53–0.60) | (0.27–0.34) |
| White-tailed deer | 1.05 | 1.11 | **1.14** | **1.31** | 1.04 | 1.09 | **0.92** | **0.85** | **0.88** | **0.77** |
| | (1.00–1.10) | (1.01–1.22) | (1.07–1.21) | (1.16–1.47) | (1.00–1.09) | (0.99–1.19) | (0.88–0.96) | (0.78–0.92) | (0.84–0.92) | (0.70–0.84) |

and human activity. These findings offer insight into how mammals might use time as a resource to adapt and persist in urban ecosystems.

We found that coyote had a greater probability of nocturnal behavior in areas with greater human densities. These findings are in agreement with past studies from single cities that documented increases in coyote nocturnal behavior in areas of higher human activity (*Gallo et al., 2019*; *Grinder and Krausman, 2001a*; *Riley et al., 2003*; *Tigas et al., 2002*). Notably, vehicular collisions are a major mortality factor for coyotes (*Grinder and Krausman, 2001b*) and coyotes have been typically persecuted by humans when they come in close contact (*Dunlap, 1988*; *Young et al., 2019*). Thus, a shift to nocturnal activity when traffic volumes are usually lower and humans are less active outdoors may be particularly important to survival in urban landscapes (*Murray and St. Clair, 2015*).

Red fox became less nocturnal as the proportion of local greenspace (i.e., available habitat) increased, a finding which may be explained by competition with coyote. Coyote and red fox exhibit a clear dominance hierarchy, whereby the dominant coyote negatively affects the subordinate red fox via competition and predation (*Gosselink et al., 2003*). Research has shown that urban coyotes occupy larger areas of greenspace (*Gehrt et al., 2009*). When more greenspace is available around a site, and presumably a higher probability of coyote presence, red foxes may become more diurnal to temporally avoid coyotes and reduce the risk of an interaction. Yet, when greenspace is limited, and presumably there is a lower probability of coyote presence, red foxes could be more active during nighttime hours with less risk of an interaction. These results may present further evidence that a shift in nocturnality of an apex predator or dominate species facilitates a 'behavioral release' in subordinate species as described by *Frey et al., 2020*.

In most cases, the human aspects of urban environments captured by our predictor variables had opposite effects on omnivores and herbivores. While human population densities increased nocturnal activity for coyote, it decreased nocturnal activity for white-tailed deer, eastern cottontail, and raccoon. Prey species may become more diurnal to avoid increased activity in nocturnal predators (*Mills and Harris, 2020*) or they may be utilizing increased human activity during daytime hours as a human-mediated shield. Prey species are known to spatially distribute themselves near human activity to act as a shield from predators (*Berger, 2007*; *Shannon et al., 2014*). In these cases, prey species may also utilize time as a human-mediated shield, exhibiting more activity at times of high human activity (daytime) in areas of high human densities. These results may seem counterintuitive given that increasing impervious cover increased the probability of nocturnal behavior exhibited by deer and raccoon (*Figure 4*) and selection for nighttime hours by striped skunk (*Figure 3*). However, a majority of impervious surfaces in the United States are roads and parking lots – places of high vehicular traffic (*Frazer, 2005*). Similar to coyote, vehicular collisions are a major source of mortality for these species (*Glista et al., 2009*). Therefore, a shift to nocturnal activity in areas with high impervious cover may be particularly important to their survival in cities and a sign of fine-scale modulation of temporal selection based on local environments.

Similarly, raccoon and white-tailed deer selected more for nighttime hours (*Figure 3*) in locations with high levels of vegetation cover. More vegetation equates to more protective cover. Therefore, we suggest that raccoons and white-tailed deer can use the same temporal habitat as their predators (i.e., coyote) – but with less risk – when there is more physical cover. On the other hand, eastern cottontail were more diurnal with increased vegetation (*Figures 3 and 4*), suggesting that more vegetation cover provides shelter from other perceived threats (i.e., humans; *Gallo et al., 2019*) and may allow eastern cottontail to select periods of high human activity (i.e., day). Interactions between various urban characteristics, which we did not examine in this study, should be further explored to fully understand how these characteristics jointly influence the temporal patterns of urban wildlife species.

Our results highlight the complexity of trade-offs for urban wildlife. In most cases, we found diverging activity patterns between coyote (a common urban apex predator) and subordinate or prey species in response to physical characteristics of urban environments. To persist in urban environments, it appears that urban species may have to modulate behaviors to contend with both anthropogenic risks and risk from predation or competition. Our results add to a growing body of literature that indicate species interactions in human-dominated landscapes may be better understood by explicitly considering the role humans play in those interactions (*Berger, 2007*; *Blecha et al., 2018*; *Gallo et al., 2019*; *Magle et al., 2014*).

We also found evidence that local climate, specifically temperature, regulated the diel behavior of many species. For example, white-tailed deer, eastern cottontail, and striped skunk became more diurnal as temperatures increased, presumably foraging more during the day in warmer seasons when more vegetation biomass is available. Virginia opossum showed a decrease in nocturnal behavior at lower temperatures. Given their poor thermoregulation abilities, poorly insulated fur, and cold-sensitive hands, ears, and tails (*Kanda, 2005*), it seems likely that Virginia opossum are morphologically constrained and thus unable to alter their diel activity patterns at colder temperatures. These results call attention to the importance of considering the impacts of morphology, physiology, and life history on a species' capacity to adapt to environmental change. Given the interacting effects of climate change and urbanization (*Stone, 2012*), future research should explore how life history traits mediate temporal distributions of species activity – particularly as cities are rapidly warming (*Oleson et al., 2013*).

We did not find changes in diel activity for some species in response to our predictor variables. These results could be due to a lack of data on a particular species (i.e., bobcat) or because we did not sample across a large enough urban–rural gradient. Remote regions were not sampled in our study design, and some species may change their behavior at a lower level of urban intensity that we did not sample. Combining datasets from more rural and remote areas (e.g., Snapshot Serengeti *Swanson et al., 2015*, Snapshot USA *Cove et al., 2021*) could allow us to identify the level of human development that elicits changes in diel activity for potentially sensitive species. Finally, our analysis was limited to the physical characteristics of cities. Additional characteristics like chronic noise, light pollution, resource supplementation, and species interactions influence animal behaviors and should be explored in future research.

A variety of methods have been developed to study animal activity patterns and temporal behavior using time-stamped camera data (see *Frey et al., 2017* and references within). However, very little work has been done to quantify changes in temporal behavior across continuous independent variables (*Cox et al., 2021*; *Gaston, 2019*). Here, we built upon *Farris et al., 2015* and developed an analytical approach to quantify temporal resource selection across continuous environmental gradients. Although we have developed a new analytical tool to measure temporal selection, a theoretical context for temporal habitat selection is needed and a further understanding of disproportional selection relative to the number of hours available is a promising avenue for future animal biology research.

Temporal partitioning may facilitate human–wildlife coexistence and effectively increase available habitats for species in cities. Temporal partitioning may also limit contact between people and animals, potentially reducing negative encounters like disease transmission and attacks on people (*Gaynor et al., 2018*). From a management perspective, ignoring diel behavior can result in biased estimates of species abundance and patterns of habitat use and lead to misinformed conservation measures (*Gaston, 2019*). Additionally, recognizing plasticity in species behavior can lead to better predictions of vulnerability to anthropogenic disturbances (*Gaynor et al., 2018*). Therefore, we recommend that diel activity and temporal partitioning be considered in conservation and management approaches.

We have shown that mammals have significant variation in the use and selection of time throughout the diel period. Additionally, our approach allowed us – for the first time – to quantify changes in diel activity across gradients of environmental change and across multiple urban areas, revealing that changes in diel patterns are influenced by natural and human landscape characteristics. Our results highlight the need to understand how a larger proportion of the animal community responds to urbanization, and provide evidence of behavioral plasticity that allows some species to adapt to and persist in human-dominated systems. Future projections of urban growth signal that urban areas will continually encroach on wildlife habitat. Therefore, it is imperative that we consider animal behavioral responses to urbanization as we plan human spaces that can also accommodate wildlife.

## Materials and methods

### Study design

The number of sampling sites per city ranged from 24 to 113 ($\bar{x}$ = 45.30, sd = 28.65). In each city, sampling sites were placed along a gradient of urbanization (high to low population density and impervious cover). At each sampling site (n = 453) we placed one Bushnell motion-triggered infrared Trophy Cam (Bushnell Corp., Overland Park, KS, USA). Sampling sites were located in greenspaces, such as city parks, cemeteries, natural areas, utility easements, and golf courses. To increase the detection probability of each species we placed one synthetic fatty acid scent lure in the camera line of sight, and lures were replaced on 2-week intervals if missing to remain consistent throughout the study. However, *Fidino et al., 2020* later found that this type of lure has little to no effect on the detectability of most urban mammals. We used observation data collected between January 2017 and December 2018. However, not all cities were sampled continuously throughout the study period (*Supplementary file 1a*).

### Data processing

For each species, we defined a single detection event as all photos taken within a 15-min period at each camera station (*Farris et al., 2015*; *Ridout and Linkie, 2009*). We categorized each detection event as either 'dawn', 'dusk', 'day', 'night', and 'deep night' using the *suncalc* package (*Thieurmel and Elmarhraoui, 2019*) in R ver 4.2.0 (*R Development Core Team, 2021*). The suncalc package defines and calculates 'dawn' as starting when morning astronomical twilight (when the center of sun is 18° below the horizon) begins and ending when the bottom edge of the sun touches the horizon. 'Dusk' was defined as the beginning of evening astronomical twilight to the point when it became dark enough for astronomical observations. 'Day' was defined as the period between dawn and dusk. We considered the nighttime as two distinct time periods (night and deep night), because some species may be nocturnal but use different hours of the night to reduce the risk of human interactions (*Gehrt et al., 2009*). We defined 'night' as the periods between the end of dusk and 1 hr before the moment when the sun is at the lowest point (astronomically darkest moment of the night), and from 1 hr after the moment when the sun is at the lowest point to dawn. 'Deep night' was therefore categorized as 1 hr before and after the moment when the sun was at the lowest point. We accounted for the date, geographical location, and daylight savings time of each detection events. Therefore, the amount of time available in each category could vary geographically and seasonally.

### Predictor variables

To assess how characteristics of urban environments influenced diel activity of urban wildlife mammals, we calculated site-level predictor variables within a fixed-radius buffer around each sampling site. Fixed-radius buffers varied in size among species and were based on the typical home range of each species: 500 m fixed-radius buffer for eastern cottontail (*Hunt et al., 2013*), Virginia opossum (*Fidino et al., 2016*; *Wright et al., 2012*), and white-tailed deer (*Etter et al., 2002*); 1 km fixed-radius buffer for striped skunk (*Weissinger et al., 2009*) and raccoon (*Rosatte, 2000*), and 1.5 km fixed-radius buffer for coyote (*Gehrt et al., 2009*; *Riley et al., 2003*), red fox (*Mueller et al., 2018*), and bobcat (*Riley et al., 2003*). In our analysis, we included variables calculated within each species' fixed-radius buffer that described two contrasting characteristics of urban ecosystems, the natural and the human-built environment (*Supplementary file 1d*). We also included average daily temperature to account for possible seasonal changes in diel activity.

*Urban features* – To characterize urbanization around each sampling site, we calculated human population density (individuals/km$^2$) and mean impervious cover (%). Population density was extracted from Block Level Housing Density data (*Radeloff et al., 2018*) created from 2010 U.S. Census data (*U.S. Census Bureau, 2010*). Mean impervious cover was calculated from the 2011 National Land Cover Database (NLCD) 30 m resolution Percent Developed Imperviousness data (*Homer, 2015*).

*Natural features* – To characterize natural features, we calculated the proportion of vegetation cover and the proportion of available greenspace (i.e., potential habitat) around each site. To calculate the proportion of vegetation cover around each sampling site, we first calculated the Normalized Difference Vegetation Index (NDVI) using U.S. Geological Survey 30 m resolution LandSat 8 data that (1) covered the entire study area of each city, (2) was taken during a summer month that coincided with the respective city's sampling period, and (3) contained less than 15% cloud cover. LandSat 8 imagery was downloaded with R using the *getSpatialData* package (*Schwalb-Willmann, 2019*). We then calculated vegetation cover as the proportion of cells within each fixed-radius buffer that had an NDVI value representing substantial vegetation cover

(>0.2; https://climatedataguide.ucar.edu/climate-data/ndvi-normalized-difference-vegetation-index-noaa-avhrr). To calculate available greenspace, we extracted the proportion of 2011 NLCD Land Cove 30 m resolution raster cells within each fixed-radius buffer that were classified as forest, shrubland, herbaceous, wetland, and developed open space (which included urban green spaces).

*Seasonality* – Because weather that defines each calendar season varies across our sampled longitudinal gradient, we used daily average temperature (i.e., mean temperature on the day of a given detection event) as a continuous covariate to describe seasonality. For each day and location of a detection event, we recorded the daily average temperature from the National Climatic Data Center using the R package *rnoaa* (*Chamberlain, 2020*). We used data from the nearest weather station to each city that recorded daily weather during our study period (*Supplementary file 1e*).

All predictor variables were group mean centered by the respective city and scaled by the global standard deviation for each variable. This scaling eases parameter interpretation and makes parameter estimates less sensitive to unequal sample size among cities (*Fidino et al., 2021*; *Milliren et al., 2018*).

## Quantifying the influence of urban characteristics on diel patterns

By splitting diel time into $k$ in 1, ..., $K$ categories where $k$ represents a single category and $K$ represents the total number of categories, we estimated the probability a detection event occurs in each category ($k$) for each species using multinomial (or softmax) regression (*Kruschke, 2011*). To do so, we let $y_i$ be the time category of the $i$th in 1, ..., $I$ detection events where $I$ is the total number of detection events. Softmax regression is similar to logistic regression (*Kruschke, 2011*), however in our case we have multiple outcomes and therefore assume $y_i$ is a categorical random variable, where $\phi$ is a probability vector of the $K$ categories $\phi = \begin{bmatrix} \phi_1 \phi_2 \phi_3 \phi_4 \phi_5 \end{bmatrix}$, $\phi_1 = 1 - \phi_2 - \phi_3 - \phi_4 - \phi_5$, and $\sum \phi = 1$ such that:

$$y_i \sim \text{Categorical}\left(\phi\right). \tag{1}$$

To understand mechanistic changes in species-specific diel activity patterns and assess the influence that each predictor variable had on the temporal activity of each species, we let $\phi_i$ be a function of covariates the softmax link function,

$$\phi_{i,k} = \frac{\exp(\lambda_{i,k})}{\sum_{k=1}^{K} \exp(\lambda_{i,k})} \tag{2}$$

where is the log-linear predictor for detection event $i$ and category k. The softmax function (*Equation 2*) states that the probability of outcome $k$ is the exponentiated linear propensity of outcome $k$ relative to the sum of the exponentiated linear propensities across all outcome of a set of categories K (*Kruschke, 2011*). We set our reference category as 'day' (i.e., $k = 1$). In our model the log-linear predictor of each outcome is then

$$\lambda_{i,k} = \begin{cases} \log\left(\alpha_{i,k}\right) & k = 1 \\ \boldsymbol{x}_j^T \boldsymbol{\beta}_{j,k} + \beta_{c[i],k} + \log\left(\alpha_{i,k}\right) & k > 1. \end{cases} \tag{3}$$

In *Equation 3*, $\boldsymbol{\beta}_{j,k}$ coefficients correspond to the effect of greenspace availability, impervious cover, vegetation cover, human population density, and daily average temperature for $k > 1$. As detection events within each city may not be wholly independent, we included a random intercept for city, $\beta_{c[i],k}$, where $\boldsymbol{c}$ is a vector of length $I$ that denotes which city detection event $i$ occurred (*Gelman and Hill, 2006*). Finally, to account for the different amount of time available to animals among the $K$ categories, we also included a log offset term, $\log(\alpha_{k,i})$, where $\alpha_{k,i}$ is the number of hours available in category $k$ at the time of detection event $i$. This form of multinomial regression is equivalent to a logistic regression model with a spatial categorical covariate with $K$ levels, where the offset accounts for varying availability. As such, our model approximates the weighted distribution used in resource selection functions assuming an exponential link (*Hooten et al., 2017*). Exponentiated coefficient estimates greater than one indicates 'selection' and less than one indicates 'avoidance', relative to the day reference category.

Because we considered 'day' ($k = 1$) as our reference outcome, we set $\beta_{c[i],1} = 0$ and $\beta_{j,1} = 0$ (*Equation 3*). The remaining $\beta_{j,k}$ parameters were given Laplace(0,$\pi$) priors as a form of categorical LASSO regularization (*Tutz et al., 2015*). The Laplace distribution shrinks values of variables with low explanatory

significance toward 0 based on the tuning parameter $\pi$; thus, reducing variability of estimates when multicollinearity exists between variables (*Oyeyemi et al., 2015*). We took a fully Bayesian approach to variable selection by estimating the hyperparameter $\pi$ (*van Erp et al., 2019*), which was given a uniform(0.001,10) prior distribution. The random effect, $\beta_{c[i],k}$, was given a Normal($\mu_k,\tau_k$) prior for each city where $\mu_k \sim$ Normal(0,10) and $\tau_k \sim$ Gamma(1,1). This approach allows for partial pooling which improves estimation and makes explicit that city-specific observations are not wholly independent (*Gelman and Hill, 2006*).

Models were fit using an Markov Chain Monte Carlo (MCMC) algorithm implemented in JAGS ver 4.2.0 (*Page et al., 2003*) using the *runjags* package (*Denwood, 2016*) in R. Fourteen parallel chains were each run from random starting values. The first 20,000 iterations from each chain were discarded and every seventh iteration was kept to reduce autocorrelation among the samples. A total of 75,000 iterations were obtained for each model. Model convergence was assessed by checking that the Gelman–Rubin diagnostic statistic for each parameter was <1.1 (*Gelman and Rubin, 1992*) and by visually inspecting the trace plots of MCMC sample.

## Acknowledgements

The authors would like to thank all the field technicians, students, and assistants associated with the Urban Wildlife Information Network for data collection and photo processing. We would also like to thank the operations, facilities, and administrative staff at our respective institutions as their work behind the scenes is vital to our research. Funding was provided by the Abra Prentice-Wilkin Foundation. We would also like to thank N Clemente, J Kimlinger, and Pariveda Solutions for their help with an application to store and tag our camera trap images. Finally, we would like to thank the two reviewers whose comments substantially improved this manuscript.

## Additional information

### Funding

| Funder | Grant reference number | Author |
|---|---|---|
| Abra Prentice-Wilkin Foundation | | Travis Gallo<br>Mason Fidino<br>Elizabeth W Lehrer<br>Maureen H Murray<br>Seth B Magle |

The funders had no role in study design, data collection, and interpretation, or the decision to submit the work for publication.

### Author contributions

Travis Gallo, Conceptualization, Data curation, Formal analysis, Investigation, Methodology, Visualization, Writing - original draft, Writing – review and editing; Mason Fidino, Conceptualization, Formal analysis, Methodology, Visualization, Writing - original draft, Writing – review and editing; Brian Gerber, Conceptualization, Methodology, Writing - original draft, Writing – review and editing; Adam A Ahlers, Julia L Angstmann, Max Amaya, Amy L Concilio, David Drake, Danielle Gay, Travis J Ryan, Carmen M Salsbury, Heather A Sander, Theodore Stankowich, Jaque Williamson, J Amy Belaire, Kelly Simon, Data curation, Writing – review and editing; Elizabeth W Lehrer, Conceptualization, Data curation, Writing – review and editing; Maureen H Murray, Colleen Cassady St Clair, Conceptualization, Writing – review and editing; Seth B Magle, Conceptualization, Data curation, Funding acquisition, Project administration, Resources, Supervision, Writing – review and editing

### Author ORCIDs

Travis Gallo http://orcid.org/0000-0003-2877-9848
Brian Gerber http://orcid.org/0000-0001-9285-9784
Travis J Ryan http://orcid.org/0000-0003-2039-5989
Kelly Simon http://orcid.org/0000-0002-9694-5548

Decision letter and Author response
Decision letter https://doi.org/10.7554/eLife.74756.sa1
Author response https://doi.org/10.7554/eLife.74756.sa2

## Additional files

### Supplementary files

• Supplementary file 1. Additional data tables that summarize observations from remotely triggered wildlife cameras and environmental variables associated with sampling sites across 10 US cities.

• Transparent reporting form

### Data availability

All related data and R scripts have been deposited at Dryad: https://doi.org/10.5061/dryad.fxpnvx0tb.

The following dataset was generated:

| Author(s) | Year | Dataset title | Dataset URL | Database and Identifier |
|---|---|---|---|---|
| Gallo T, Fidino M, Gerber B, Ahlers A, Angstmann J, Amaya M, Concilio A, Drake D, Gray D, Lehrer E, Murray M, Ryan TJ, Salsbury C, Sander H, Stankowich T, Williamson J, Belaire A, Simone K, Magle S, St Clair C | 2022 | Mammals adjust diel activity across gradients of urbanization | https://doi.org/10.5061/dryad.fxpnvx0tb | Dryad Digital Repository, 10.5061/dryad.fxpnvx0tb |

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

# Appendix 1

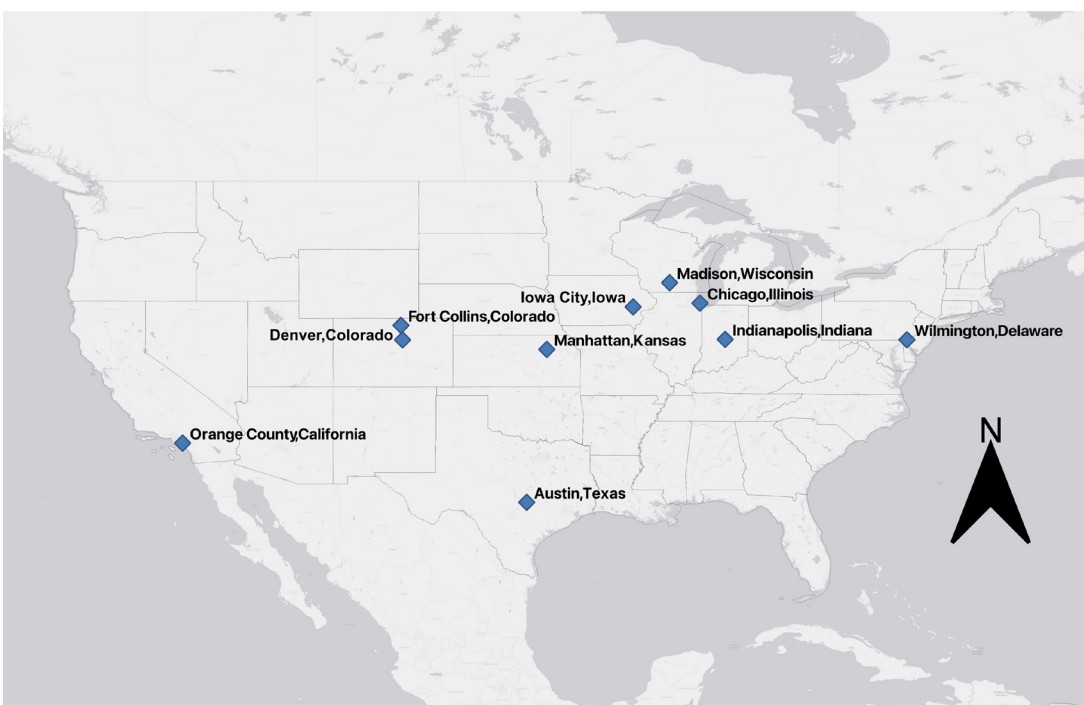

**Appendix 1—figure 1.** US cities where remotely triggered wildlife cameras were deployed to assess diel patterns in urban mammals.

