## [Editor Report]

This study will be of interest to wildlife ecologists and conservation practitioners. The authors took a collaborative approach and collated a large dataset of wildlife camera trap recordings across cities in the USA. The analyses reveal variability in diel activity among species and cities, providing important insights into the effects of urbanization.

---

## [Decision Letter]

**Decision letter after peer review:**

Thank you for submitting your article "Mammals adjust diel activity across gradients of urbanization" for consideration by *eLife*. Your article has been reviewed by two peer reviewers, and the evaluation has been overseen by Yuuki Watanabe as the Reviewing Editor and Christian Rutz as the Senior Editor. The following individuals involved in the review of your submission have agreed to reveal their identity: Daniel Cox (Reviewer #1); Jason T Fisher (Reviewer #2).

The reviewers have discussed their reviews with one another, and the Reviewing Editor has drafted this decision letter to help you prepare a revised submission.

Essential revisions:

Both reviewers agreed that this is an interesting study, and their comments are mostly suggestions for better presentation and enhanced clarity regarding the methods. You are encouraged to refine the manuscript as much as possible by considering the comments provided in the original reports below.

*Reviewer #1 (Recommendations for the authors):*

The introductory paragraph could do with tightening to fully capture the broad range of different time partitioning strategies.

L58: Persist is the wrong term here. Many species positively thrive in urban areas and do better than they do in the wider countryside.

L66-68: Rather than limit this to the two examples given, I feel it would be better to give the broad range of external factors that can cause species to shift their activity e.g. predator-prey relationship, thermoregulation, food availability, local climatic conditions, high seasonal variability or unpredictability, lunar cycles, competition, apex predators, etc.

L100-101: Carnivores can also be thought of as a special case with their diel niches because they are inherently flexible in their activity and many have a 'cathemeral eye'.

L112-114: Before reaching the rest of the manuscript, it would be helpful here to further clarify the differences between objectives 1 and 2. Possibly give an example of the changing behavior that is expected?

L131: It would be helpful to provide a link to the website for the 'urban wildlife network' at the first mention, because many readers will not have heard of it.

L132-135: For the non-US readers, it would be helpful to include a map in the SI showing the distribution of the cities. I would also suggest, either removing the states or giving the full spelling.

L155: I am not sure what the 'softmax function' is at this stage of the manuscript. I would suggest either clarifying or removing and leaving the explanation to the methods.

L213-214: Figure 3g suggests that eastern cottontails were more likely to select diurnal hours as temperatures increase? As shown in Figure 2 and 3.

L265: Should you be citing phylopic www.phylopic.org here or equivalent?

L322-333: Rather than becoming more diurnal from human pressures, it may be that the primary driver of increased diurnal activity is to avoid increased activity in nocturnal predators (see Mills and Harris 2020).

L427: For readers who don't know, they might be interested to know that astronomical sunrise and set is when the sun goes above -18 degree.

L435: I think that the term 'darkest hours' is misleading. Artificial light at night is prevalent throughout urban areas, with strong levels of skyglow even if there are no direct sources close to the camera traps. It would be better to rename this period along the lines of the 'quietest hours', because mammal diel activity is to do with the lack of human activity as opposed to illumination.

On this topic it is a shame that no data were available that measured ALAN at each site. ALAN allows species with high visual acuity (a diurnal adaptation) to operate at night. It would have been interesting to test whether this was a dominant effect driving activity patterns.

The following reference might be useful https://onlinelibrary.wiley.com/doi/full/10.1111/ecog.05251

L443-448: Nicely done.

L450: I assume you mean average daily temperature? Is this not just a repetition of what comes below?

L453: Possibly just to clarify, add 'within each species fixed radius buffer.'

L460: Are these two nature feature variables not highly correlated?

L483: I have some experience of non-Bayesian multinomial modelling and can appreciate the difficulty of explaining it. I think much of the below can be clarified with a careful explanation here of what the different parts are. The authors assume too much and need to spell out more clearly how the categorical model works.

So, there are 5 K categories (dark night, night, dawn, dusk, day).

Why k and K?

k in 1?

'…of the ith in 1,….' What does this mean?

Why a categorical 'random' variable?

1⋅ ɸ = 1 – is this shorthand for stating that the sum of probabilities across the 5 diel activities is equal to 1?

L492: First, mention of the softmax function in the methods. Is this function equation 2? Further, an explanation is required to state what a softmax function is and what it does.

L496: Are k and K the same thing? This would suggest they are?

How does the model control for zero-inflated data (i.e., all the camera traps where the species was not recorded)?

L500: How was the random intercept for each city obtained? Can this not be controlled for by simply including the city as a random factor in the model?

Possibly also expand the explanation into the SI?

L519: As a general point for readers wanting to repeat this analysis, which they might well do given the increased interest in activity patterns, throughout the paper the individual functions should be given along with the R packages.

Figure 3: It might look good if you could also add the silhouettes in Figure 4 to the top right of each panel in Figure 3 and Figure 2?

Table S2: Why not give full state names in the table heading?

Table S3: It would be clearer if full city names are also given.

Dryad link on the manuscript does not work, and nothing comes up when the paper title is searched for within Dryad.

Literature cited:

Mills and Harris (2020) Human disrupt access to prey for large African carnivore. *eLife*, 9, e60690.

*Reviewer #2 (Recommendations for the authors):*

Introduction: I mentioned in the public review that your background on using camera traps to analyze diel activity is a little sparse. A review is available, which also highlights some challenges the authors have not apparently considered in the paper. It's not necessary to cite it (it is my paper) but the content may prove useful (Frey et al., 2017).

Line 84: Again it's not necessary to cite this paper (as it's mine) but Frey et al., (2020) refute the simplistic patterns shown in Gaynor et al., (2018) and demonstrate that species adjust their diel cycle according to multiple perceived risks – cascading from top predators to mesopredators. I see you suggest this as a mechanism for foxes/coyotes in Lines 310-381; might be good to lend some support to your argument.

Lines 223-225: I did not understand this logic – why did you combine night/darkest night because dusk/dawn probabilities were low?

Line 302: Formatting of reference "M.I. Grinder and Krausman".

In my version, Figure 2 is quite small.

Frey, S., J. T. Fisher, A. C. Burton, and J. P. Volpe. 2017. Investigating animal activity patterns and temporal niche partitioning using camera‐trap data: Challenges and opportunities. Remote Sensing in Ecology and Conservation 3:123-132.

Frey, S., J. Volpe, N. Heim, J. Paczkowski, and J. Fisher. 2020. Move to nocturnality not a universal trend in carnivore species on disturbed landscapes. Oikos 129:1128-1140.

Ridout, M. S., and M. Linkie. 2009. Estimating overlap of daily activity patterns from camera trap data. Journal of Agricultural, Biological, and Environmental Statistics 14:322-337.

---

## [Author Response]

Reviewer #1 (Recommendations for the authors):The introductory paragraph could do with tightening to fully capture the broad range of different time partitioning strategies.L58: Persist is the wrong term here. Many species positively thrive in urban areas and do better than they do in the wider countryside.

To recognize that some species thrive in urban environments, the sentence now reads:

“Our results highlight the complexity with which temporal activity patterns interact with local environmental characteristics, and suggest that urban mammals may use time along the 24-hour cycle to reduce risk, adapt, and therefore persist, and in some cases thrive, in human-dominated ecosystems.” (Lines 59)

L66-68: Rather than limit this to the two examples given, I feel it would be better to give the broad range of external factors that can cause species to shift their activity e.g. predator-prey relationship, thermoregulation, food availability, local climatic conditions, high seasonal variability or unpredictability, lunar cycles, competition, apex predators, etc.

We have included these added examples and citations to support them. The sentence now reads:

“For example, some species make fine-scale adjustments to their temporal behavior to respond to predation risk (van der Vinne et al., 2019), competition (Kronfeld-Schor and Dayan, 2003), food availability (Owen-Smith, 2008), seasonal variability in local climatic conditions (Maloney et al., 2005), and even lunar cycles (Prugh and Golden, 2014).” Lines (66-70).

L100-101: Carnivores can also be thought of as a special case with their diel niches because they are inherently flexible in their activity and many have a 'cathemeral eye'.

This is a good point and the reason for the following sentence (Lines 108-109) highlighting that not all species have morphologies that allow them that flexibility

L112-114: Before reaching the rest of the manuscript, it would be helpful here to further clarify the differences between objectives 1 and 2. Possibly give an example of the changing behavior that is expected?

We recognize now that these two objectives could have been summarized into a more concise single objective. Thank you for pointing this out. The sentence now reads:

“Our research objective was to determine which species change their diel activity across gradients of urbanization and identify what characteristics of the urban environments have the strongest association with changes in diel activity.” (Lines 117-119)

L131: It would be helpful to provide a link to the website for the 'urban wildlife network' at the first mention, because many readers will not have heard of it.

Website added (Line 136)

L132-135: For the non-US readers, it would be helpful to include a map in the SI showing the distribution of the cities. I would also suggest, either removing the states or giving the full spelling.

We have added a map as an appendix (Appendix 1 – Figure 1) and included the full state names throughout the manuscript and supplemental materials. Thank you for this suggestion.

L155: I am not sure what the 'softmax function' is at this stage of the manuscript. I would suggest either clarifying or removing and leaving the explanation to the methods.

We reordered the sentence to describe the procedure before the tool to improve clarity. However, we retain the term softmax as this is the appropriate term for the procedure that converts the results of a multinomial model to the probability scale. The sentence (Lines 159-162) now read:

“We also estimated the influence that each predictor variable had on the probability of activity in each time category, including the ‘day’ category, using the softmax function – a generalization of the inverse logit link for >2 modeled categories (Kruschke, 2011).”

We also added more details in the methods section in response to a similar comment below.

L213-214: Figure 3g suggests that eastern cottontails were more likely to select diurnal hours as temperatures increase? As shown in Figure 2 and 3.

Thank you for catching this error. We have changed this in the results (Lines 223-225XX). We were interpreting this result correctly in the Discussion, so we did not make any changes in that portion of the manuscript.

L265: Should you be citing phylopic www.phylopic.org here or equivalent?

All of our silhouettes were obtained from the Google Image search engine with the usage rights set to Creative Commons license. We did, however, check the Phylopic database and found that only the bobcat image is contained in the database. The licensing for this image is set to Public Domain Dedication 1.0 and:

“The person who associated a work with this deed has dedicated the work to the public domain by waiving all of his or her rights to the work worldwide under copyright law, including all related and neighboring rights, to the extent allowed by law.”

L322-333: Rather than becoming more diurnal from human pressures, it may be that the primary driver of increased diurnal activity is to avoid increased activity in nocturnal predators (see Mills and Harris 2020).

We have included this additional explanation and included the Mills and Harris citation. Thank you for sharing the reference with us. The sentence (Lines 331-333) now reads:

“Prey species may become more diurnal to avoid increased activity in nocturnal predators (Mills and Harris, 2020) or they may be utilizing increased human activity during daytime hours as a human-mediated shield.”

L427: For readers who don't know, they might be interested to know that astronomical sunrise and set is when the sun goes above -18 degree.

We have added this tidbit of information (Line 442). Thanks!

L435: I think that the term 'darkest hours' is misleading. Artificial light at night is prevalent throughout urban areas, with strong levels of skyglow even if there are no direct sources close to the camera traps. It would be better to rename this period along the lines of the 'quietest hours', because mammal diel activity is to do with the lack of human activity as opposed to illumination.

This is a good point, however, the idea that mammal diel activity correlated to human activity was our main hypothesis, and therefore we did not want to pre-emptively claim this to be the ‘quietest part of night’. Additionally, the categorizing of this time period had nothing to do with people, and everything to do with the sun being at its lowest point (Line 450) and thus the darkest moment of the night. We recognize the reviewers point that without measuring light at each of our 200+ sites we can’t for sure say this is the darkest period. Therefore, we have chosen to change this category to “Deep Night” throughout the manuscript.

We also rephrased this portion of the methods to read:

“We considered the nighttime as two distinct time periods (night and deep night), because some species may be nocturnal but use different hours of the night to reduce the risk of human interactions (Gehrt et al., 2009). We defined ‘night’ as the periods between the end of dusk and one hour before the moment when the sun is at the lowest point (astronomically darkest moment of the night), and from one hour after the moment when the sun is at the lowest point to dawn. ‘Deep Night’ was therefore categorized as one hour before and after the moment when the sun was at the lowest point. We accounted for the date, geographical location, and daylight savings time of each detection events. Therefore, the amount of time available in each category could vary geographically and seasonally.” (Lines 445-453)

On this topic it is a shame that no data were available that measured ALAN at each site. ALAN allows species with high visual acuity (a diurnal adaptation) to operate at night. It would have been interesting to test whether this was a dominant effect driving activity patterns.The following reference might be useful https://onlinelibrary.wiley.com/doi/full/10.1111/ecog.05251

We 100% agree that ALAN is something important to explore. However, our research group has tried to use the VIIRS dataset (1km resolution) and the World Atlas of the artificial night sky brightness (~2km resolution; http://www.inquinamentoluminoso.it/cinzano/download/0108052.pdf) on past projects and have found that at 1km resolution or greater, we have very little variation across our more urban sites. Therefore, we have been unable to use these datasets to model animal activity or distributions. We have also worked with the Cities at Night project (https://citiesatnight.org/), but do not have coverage for all of our cities. We do acknowledge that light pollution is an important avenue for future research on Lines 386-388.

L443-448: Nicely done.

Thank you

L450: I assume you mean average daily temperature? Is this not just a repetition of what comes below?

We rephrased to say “average daily temperature” to improve clarity (Line XX). Our intention for this paragraph (Line 466) was to introduce the readers to our predictor variables with a justification for each. Then to follow with more details about how they were calculated. So, yes, in a way it is repetitive to below, but we feel it helps with the flow.

L453: Possibly just to clarify, add 'within each species fixed radius buffer.'

This sentence now reads:

“In our analysis, we included variables calculated within each species’ fixed-radius buffer that described two contrasting characteristics of urban ecosystems, the natural and the human-built environment (Supplementary file 1d).” (Lines 463-465)

L460: Are these two nature feature variables not highly correlated?

Yes, they were correlated, but our LASSO regularization allows for models to contain correlated variables. We have added more context in our methods to describe how this approach is appropriate when you have collinearity between variables.

Lines 537-542: The remaining β_*j,k*_ parameters were given Laplace(0,π) priors as a form of categorical LASSO regularization (Tutz et al., 2015). The Laplace distribution shrinks values of variables with low explanatory significance toward 0 based on the tuning parameter π; thus, reducing variability of estimates when multicollinearity exists between variables (Oyeyemi et al., 2015). We took a fully Bayesian approach to variable selection by estimating the hyperparameter π (van Erp et al., 2019), π which was given a uniform(0.001,10) prior distribution.

L483: I have some experience of non-Bayesian multinomial modelling and can appreciate the difficulty of explaining it. I think much of the below can be clarified with a careful explanation here of what the different parts are. The authors assume too much and need to spell out more clearly how the categorical model works.So, there are 5 K categories (dark night, night, dawn, dusk, day).Why k and K?k in 1?'…of the ith in 1,….' What does this mean?

This is appropriate math notation. *k* is an individual category and K is the total number of categories. Therefore, we are describing that we conducted the analysis from the first k (k = 1) to the total number of K categories. We have added more information into this portion to improve clarity for both *k* and *i*.

Lines 503-506: By splitting diel time into *k* in 1,…,K categories where *k* represents a single category and K represents the total number of categories, we estimated the probability a detection event occurs in each category (*k*) for each species using multinomial (or softmax) regression (Kruschke, 2011).

Why a categorical 'random' variable?

Logistic regression is in fact a special case of softmax regression (See Kruschke 2001), except you have several outcomes. Instead of using a Bernoulli distribution as you would in logistic regression, in our case our response variable is distributed categorically. When you choose to model data with a given statistical distribution, this is the assumption you end up making. For example, if you were doing logistic regression you would assume your response variable is a Bernoulli random variable. With Poisson regression, your count data would be a Poisson random variable. These data are a categorical random variable and we are being precise about what the structure of our response variable is, which is important.

We do not think we need to add this much detail, but we have included more information and justification for the Categorical distribution on Lines 503-512-XX. This portion now reads:

“By splitting diel time into *k* in 1,…,K categories where *k* represents a single category and K represents the total number of categories, we estimated the probability a detection event occurs in each category (*k*) for each species using multinomial (or softmax) regression (Kruschke, 2011). To do so, we let *y_i_* be the time category of the *i*^th^ in 1,…,*I* detection events where I is the total number of detection events. Softmax regression is similar to logistic regression (Kruschke, 2011), however in our case we have multiple outcomes and therefore assume *y_i_* is a Categorical random variable, where ϕ is a probability vector of the K categories ϕ=[ϕ1ϕ2ϕ3ϕ4ϕ5], ϕ1=1−ϕ2−ϕ3−ϕ4−ϕ5, and ∑ϕ= 1 such that:

yi ∼ Categorical(ϕ)”

1· ϕ = 1 – is this shorthand for stating that the sum of probabilities across the 5 diel activities is equal to 1?

Yes, but we have changed the notation to the sum symbol to improve clarity (Line 510)

L492: First, mention of the softmax function in the methods. Is this function equation 2? Further, an explanation is required to state what a softmax function is and what it does.

We have included more information about what the softmax link function does on Lines 512-521. This portion now reads:

“To understand mechanistic changes in species-specific diel activity patterns and assess the influence that each predictor variable had on the temporal activity of each species, we let ϕi be a function of covariates the softmax link function,

ϕi,k= exp(λi,k)∑k=1Kexp(λi,k)where λi,k is the log-linear predictor for detection event *i* and category *k*. The softmax function (Equation 2) states that the probability of outcome *k* is the exponentiated linear propensity of outcome *k* relative to the sum of the exponentiated linear propensities across all outcome of a set of categories K (Kruschke, 2011). We set our reference category as ‘day’ (i.e., *k* = 1). In our model the log-linear predictor of each outcome is then…”

L496: Are k and K the same thing? This would suggest they are?

Yes, we believe we have addressed this with the above response and changes to the manuscript.

How does the model control for zero-inflated data (i.e., all the camera traps where the species was not recorded)?

The temporal RSF that we formulated does not consider detectability. We assumed that after 28+ days of sampling each season, we had high likelihood of detecting this group of common species given they were present. Thus, if a camera site never detected a particular species over the course of the study that site was not included in the analysis for that particular species. However, see https://www.biorxiv.org/content/10.1101/2021.06.30.450589v2.abstract for an example of extending our temporal RSF to consider detectability, which is currently in press at American Naturalist.

L500: How was the random intercept for each city obtained? Can this not be controlled for by simply including the city as a random factor in the model?Possibly also expand the explanation into the SI?

We included a random effect by indexing our intercept as follows: β*_c[i],k_* This nested indexing allowed us to estimate a separate intercept for each city and each time category (thus a random effect). Each city’s intercept was drawn from a Normal prior (Normal(μ*_k_*,τ*_k_*)) where where μ*_k_* ~ Normal(0,10) and τ*_k_* ~ Γ(1,1). This approach allows for partial pooling which approves estimation and makes explicit that city-specific observations are not wholly independent (Gelman and Hill, 2006). We have included the following statement and citation on Line 542-545 to improve clarity:

“This approach allows for partial pooling which approves estimation and makes explicit that city-specific observations are not wholly independent (Gelman and Hill, 2006).”

Yes, we could have included the city as a categorical variable in the model, but our formulation allows us to leverage the flexibility of Bayesian hierarchical models to estimate both the average across cities and city-specific estimates (including uncertainty for both) and lets us obtain results seen in Figure 1 (city specific results) and 4 (averaged responses).

L519: As a general point for readers wanting to repeat this analysis, which they might well do given the increased interest in activity patterns, throughout the paper the individual functions should be given along with the R packages.

We have included R scripts that can conduct this analysis and make the methods fully reproducible. However, we recognize the reviewer did not have access to these scripts due to an issue with Dryad. We used dozens if not 100+ functions throughout our analysis, so do not think it is necessary to include them all in the main text when a reader will have access to the R scripts. Additionally, R functions change and change names over time, and therefore listing them out in the main text does not increase reproducibility.

See: https://besjournals.onlinelibrary.wiley.com/doi/full/10.1111/2041-210X.13105

Figure 3: It might look good if you could also add the silhouettes in Figure 4 to the top right of each panel in Figure 3 and Figure 2?

We took this suggestion to heart and tried, but to get the silhouette images in with the title we had to increase the margins and it added a lot of white space to our graphs. It also made the figures as a whole too ‘busy’

Table S2: Why not give full state names in the table heading?

We have included the full state names in the table heading

Table S3: It would be clearer if full city names are also given.

We have changed the abbreviations to the full city name.

Dryad link on the manuscript does not work, and nothing comes up when the paper title is searched for within Dryad.

We will look into this.

Reviewer #2 (Recommendations for the authors):Introduction: I mentioned in the public review that your background on using camera traps to analyze diel activity is a little sparse. A review is available, which also highlights some challenges the authors have not apparently considered in the paper. It's not necessary to cite it (it is my paper) but the content may prove useful (Frey et al., 2017).

Thank you for this reference. I am slightly embarrassed that we did not come across it in our initial literature review. See the next comment for how we incorporated it into the revised manuscript.

Line 84: Again it's not necessary to cite this paper (as it's mine) but Frey et al., (2020) refute the simplistic patterns shown in Gaynor et al., (2018) and demonstrate that species adjust their diel cycle according to multiple perceived risks – cascading from top predators to mesopredators. I see you suggest this as a mechanism for foxes/coyotes in Lines 310-381; might be good to lend some support to your argument.

Thank you for this reference! We have added this information into the Introduction (Lines 83-88) to highlight that behavioral shifts towards nocturnality, in response to human disturbance, are likely context specific. The passage now reads:

“A recent global meta-analysis suggests that mammals become more nocturnal in areas with greater human disturbance (Gaynor et al., 2018). However, Frey et al., (2020) found that shifts in temporal behavior of apex predators – in response to human disturbance – caused cascading behavioral responses among meso-carnivores creating a “behavioral release”. These results highlight that temporal shifts towards nocturnality are not universal and may be context specific. “

We also included this citation and context in our paragraph about foxes and coyotes. That paragraph now reads (Lines 316-326):

“Red fox became less nocturnal as the proportion of local greenspace (i.e., available habitat) increased, a finding which may be explained by competition with coyote. Coyote and red fox exhibit a clear dominance hierarchy, whereby the dominant coyote negatively affects the subordinate red fox via competition and predation (Gosselink et al., 2003). Research has shown that urban coyotes occupy larger areas of greenspace (Gehrt et al., 2009). When more greenspace is available around a site, and presumably a higher probability of coyote presence, red foxes may become more diurnal to temporally avoid coyotes and reduce the risk of an interaction. Yet, when greenspace is limited, and presumably there is a lower probability of coyote presence, red foxes could be more active during nighttime hours with less risk of an interaction. These results may present further evidence that a shift in nocturnality of an apex predator or dominate species facilitates a “behavioral release” in subordinate species as described by Frey et al., (2020).”

Lines 223-225: I did not understand this logic – why did you combine night/darkest night because dusk/dawn probabilities were low?

To improve clarity we have rephrased this sentence to read:

“To further assess the probability of a species shifting towards nocturnality as a response to natural and anthropogenic features of the environment, we combined the probability of activity during night and deep night and predicted these values across each continuous covariates using the model results for each species.” (Lines 228-231)

Line 302: Formatting of reference "M.I. Grinder and Krausman".

We have fixed this reference

In my version, Figure 2 is quite small.

It is currently sized to be 1.5 columns for the manuscript. I am not sure why it is small, but I will make sure it is appropriately sized in the production process if accepted.